# Mixed Solvents in Multilayer Ceramic Capacitors (MLCC) Electronic Paste and Their Effects on the Properties of Organic Vehicle

**DOI:** 10.3390/polym14040685

**Published:** 2022-02-11

**Authors:** Ruolong Gan, Junrong Li, Xiuhua Cao, Jun Huang, Liying Qian

**Affiliations:** 1School of Light Industry and Engineering, South China University of Technology, Guangzhou 510640, China; 202021028911@mail.scut.edu.cn; 2State Key Laboratory of Advanced Materials and Electronic Components, Zhaoqing 526020, China; caoxh@china-fenghua.com (X.C.); squallhj1234@163.com (J.H.)

**Keywords:** electronic paste, organic vehicle, mixed solvents, solubility parameter, low residual rate

## Abstract

The copper end paste used in multilayer ceramic capacitors sintered in nitrogen atmosphere leads to carbon residues of organic vehicles, which leads to a reduction in electrode conductivity and high scrap rate. With an attempt to leave no residue in the sintering, the compatibility of solvents and thickeners should be improved because it has an important influence on the hierarchical volatilization and carbon residue of organic vehicles. In this work, the volatility of different solvents was compared, and several solvents were mixed in a definite proportion to prepare an organic vehicle with polyacrylate resins. The hierarchical volatility and solubility parameters of mixed solvents were effectively adjusted by changing proportions of different components. The thermogravimetric curves of resins and organic vehicles were measured by thermogravimetric analyzer, and the effect of solubility parameter on the dissolvability of resins in the solvent and the residual of organic vehicles were studied. Results showed that the hierarchical volatilization of solvents can be obtained by mixing different solvents; the intrinsic viscosity of the organic vehicle is higher, and the thermal decomposition residue of polyacrylate resins is lower when the solubility parameters of mixed solvents and polyacrylate resins are closer. The low residual sintering of organic vehicles can be achieved by using a mixed solvent with hierarchical volatility and approximate solubility parameters as resins.

## 1. Introduction

As one of the three basic components, the multilayer ceramic capacitor (MLCC) has the advantages of small size, high frequency, and low cost compared with other components; therefore, MLCC is widely used in various electronic products and is gradually replacing traditional aluminum electrolytic capacitors, which is one of the most widely used capacitors. MLCC is widely used in electronic equipment such as radio and television, mobile communication, and measuring instruments. With the continuous upgrading of electronic products such as smartphones, and the popularization of new communication technologies, MLCC has a huge demand in the market. With the increasing demand for MLCC in the market, the requirements of electronic information products for MLCC tend to be high frequency, low power consumption, miniaturization, superior energy storage, and low cost [1]. Copper end electronic paste is composed of three main components: conductive phase (copper), bonding phase (glass or oxide crystals), and organic vehicle. The organic vehicle consists of an organic solvent and thickener, in which the organic solvent accounts for 65–98%. In addition, polyacrylate resin is commonly used as the thickener in the organic carrier, and the sintering residue of the organic vehicle mainly depends on the thermal degradation of the polyacrylate resin thickener in the solvent. Cu electrodes with better conductivity have replaced precious-metal electrodes as the mainstream electrode material [2]; however, due to the oxidation characteristics of Cu, sintering can only be performed in nitrogen. Carbon residue caused by sintering Cu end slurry in a nitrogen atmosphere poses a great challenge to the selection of the organic vehicle composition of base metal inner electrode multilayer ceramic capacitors (BME-MLCC) [3]. The residual rate of thickener in organic vehicles is the key factor to determine the conductivity of electrodes [4]. The remaining carbon in the conductive film greatly affects the electroconductibility of the film. There are copious studies on the effects of sintering atmosphere, temperature, rate, and time on electroconductibility [5,6,7,8,9,10], but the effect of thermal degradation characteristics of a thickener on carbon residue is still not clear, and the thickener is the main component of the organic vehicle that applied in MLCC. Therefore, it is of practical significance to develop methods to select a mixed solvent with hierarchical volatility and to prepare organic vehicles with extremely low residue on the basis of the relationship between the mutual solubility of mixed solvents and thickeners.

An organic vehicle is generally composed of solvent, thickener, surfactant, thixotropic agent, and other additives. The organic solvent is the primary component of the organic vehicle, with about 65–98% of the total mass of the organic vehicle. It should dissolve the thickener quickly with excellent solubility and have a high boiling point to avoid volatilization in preparing the organic vehicle. The organic vehicle with mixed solvent can adjust the volatilization characteristics of solvents, and achieve hierarchical volatilization leading to nonvolatilization at low temperatures, and continuous and rapid volatilization at drying temperatures [11]. Meanwhile, it can improve the viscosity and fluidity of slurry, and enhance its stability and printability. In addition, the reasonable selection of mixed solvents can enhance thickener solubility and the wettability of copper, which greatly improves the performance of electronic paste in MLCC.

In this paper, according to the volatilization characterizes of different solvents, mixed solvents with different components were prepared, and the key factors of the hierarchical volatilization of organic vehicles were explored. In order to investigate the wettability of copper powder and glass powder in mixed solvents, contact-angle data were measured by a contact-angle tester. According to the solubility parameter close principle, solubility parameters of mixed solvents and polyacrylate resins were calculated and compared, while the intrinsic viscosity of polyacrylate resins in the mixed solvents was measured, and the relationship between solubility parameters and intrinsic viscosity of the organic vehicles was clarified. Furthermore, the thermal properties and residual rate of organic vehicles were characterized by TG, and the effect of compatibility between the mixed solvent and polyacrylate resins on the residue of organic vehicle was analyzed.

## 2. Experiments

### 2.1. Materials

Terpineol (T) and N, N-dimethylformamide (DMF) were purchased from Tianjin Damao Chemical Reagent Factory, China. Diethylene glycol butyl ether (DGBE), diethylene glycol butyl ether acetate (DBAC), and ethylene glycol ethyl ether acetate (CAC) were purchased from Macklin, China. All reagents were analytical-grade without further treatment before use. Copper powder and glass powder were provided by Guangdong Fenghua High Tech Co., Ltd., Fenghua, China. Polyacrylate resins A (DEGALAN^®^LP65/11, Mw ≈ 30,000 g/mol) and polyacrylate resins B (DEGALAN^®^LP65/12, Mw ≈ 44,000 g/mol) were purchased from Degussa, Germany. Monomers of polyacrylate resin were methyl methacrylate (MMA) and butyl methacrylate (BMA).

### 2.2. Characterization

#### 2.2.1. Constant Temperature Volatility of Solvent

The constant temperature volatilization test of the solvent was carried out on a collector-type magnetic heating agitator (DF-101S, Shandong Kecheng Scientific Instrument Co., Ltd., Heze, Shandong, China). We weighed 20 g each of five pure solvents or mixed solvents into a 50 mL beaker and they were marked as *m*_1_. When the oil bath reached 70, 90, 110, 130, 150, 170, and 190 °C, the beaker was placed inside it, and the temperature was maintained constant for 15 min. Then, the breaker was taken out and cooled to indoor temperature, weighed, and marked as *m*_2_. Volatilization was calculated according to Formula (1), and results were calculated by averaging the three repeated experiments. T was used as the main solvent to prepare the mixed solvent with the four other solvents at a mass ratio of 50:50, marked as T-XX, where T represents terpineol, and XX is the abbreviation of another solvent. For example, T-DMF represents the mixed solvent prepared with terpineol and DMF at a mass ratio of 50:50.
(1)$$\mathrm{Volatilization}=\frac{m_{1}-m_{2}}{m_{0}}\times 100\%,\;$$
where *m*_0_ is the actual weight of the solvent.

#### 2.2.2. Wettability of Copper Powder and Glass Powder to Mixed Solvents

After the copper powder or glass powder had been pressed into a sheet by a desktop electric tablet press (FYD-20, SCJS, Tianjin, China), the static contact angle of the copper or glass powder sheet to the mixed solvent was tested by contact-angle tester (Attentions Theta Flex, Biolin Scientific, Gothenburg, Sweden). The measurement was repeated twice to ensure reproducibility.

#### 2.2.3. Intrinsic Viscosity of Polyacrylate Resins in the Mixed Solvents

The diluted Ubbelohde viscometer (1834B, Zonwon, Hangzhou, China) was placed vertically in a high-precision thermostatic bath (VT2, ZONWON, Hangzhou, China) at 30 ± 0.01 °C, outflow time of the solvent was marked as t_0,_ and t_1_ of the solution of 1.00% polyacrylate resins A or B (10 mL); 5 mL of solvent was added into the viscometer for dilution each time, the concentration of the solution decreased to 0.67%, 0.50%, 0.40%, or 0.33%, and the outflow time of the diluted solution were recorded as t_2_, t_3,_ t_4_, and t_5_, respectively. The measurement was repeated three times to obtain the average. The intrinsic viscosity (*η*) was obtained by extrapolation.

#### 2.2.4. Thermogravimetric Analysis (TGA)

TGA was carried out on a thermogravimetry analyzer (TG209F1, Netzsch, Germany). Terpineol, DMF, and polyacrylate resins A were weighed according to the mass ratio (at 35:35:30) to prepare the organic vehicle recorded as PA-T-DMF, where PA represents polyacrylate resins A, and T represents terpineol. In the same way, the sample number of organic vehicles composed of mixed solvents and polyacrylate resins B was marked as PB-T-XX. Then, 10 mg of the organic vehicles was produced by thermogravimetry in a nitrogen atmosphere with a temperature range of 40–600 °C, heating rate of 20 °C/min. The mass change (Δ*m%*) was then recorded in the decomposition stage of polyacrylate resins in the thermal decomposition curve.

## 3. Results and Discussion

### 3.1. Constant Temperature Volatility of Solvent

The volatilization characteristics of organic vehicles are closely related to the manufacturing process of thick-film electronic components, which play an important role in the printability, drying, and sintering process of electronic pastes [11]. The organic vehicle should not be volatile at room temperature, which is conducive to the storage stability of the slurry. In the drying temperature period, the slurry should volatilize rapidly, which can shorten the processing time. Therefore, the organic vehicle should have a hierarchical volatilization curve [12]. The organic solvent is an important part of preparing organic vehicles, which is to dissolve thickeners and adjust the slurry. In the sintering process, if the organic vehicle volatilizes too slowly, it results in sagging at the edge of the end electrode because of the decrease in slurry viscosity at high temperature; if the organic vehicle volatilizes too fast, it leads to holes in the film because of a large amount of escaped gas. The electrical properties of MLCC are closely linked to film quality. Therefore, a variety of solvents with different boiling points are often used to form a mixed solvent with a hierarchical volatilization curve. The thickener should have commendable solubility in mixed solvents, which has the characteristics of nonvolatility at low temperatures and rapid hierarchical volatilization during drying.

The volatilization properties of pure and mixed solvents with a mass ratio of 50:50 are shown in Figure 1. The volatilization amount of all solvents increases continuously with the increase in temperature in the overall trend. The evaporation rates of pure solvents from high to low are DMF, CAC, T, DGBE, and DBAC, which corresponded to their boiling points at 152.8, 156, 217, 227, and 246.4 °C, respectively. This is in line with the law of easy volatilization with a low boiling point. The volatilization amounts of DMF and CAC with low boiling points were more than half at 150 °C, and they were almost completely volatilized at 190 °C. The volatilizations of DGBE and DBAC with high boiling points were indubitably small at 150 °C; although volatilizations were increased to 3 times at 190 °C, volatilizations below 40% were too small to meet the satisfactory sintering process of organic vehicles. In contrast, T volatilizes slowly at low temperatures and rapidly at 150–190 °C; volatilization at 190 °C was nearly twice that at 150 °C, which shows the excellent characteristics of not easy volatilization at low temperatures and rapid volatilization at high temperatures. Therefore, the use of T as a main component in the mixed solvents can effectively avoid problems such as film cavities and sagging after sintering. The volatilizations of T-DMF and T-CAC were lower than those of DMF and CAC because of the relatively lower volatilization ability of T in the mixed solvents. T, on the other hand, had better volatilization ability than that of DGBE and DBAC, which led to a higher volatilization amount of T-DGBE and T-DBAC.

An effective method to adjust volatilization is by combining solvents with different boiling points. This can be attributed to the partial pressure of the mixed solvent. According to Henry’s law [13], under a certain temperature and equilibrium state, the relative content of the mixed solvent determines the relative ratio of vapor pressure. The ratios of the four mixed solvents are consistent before thermal volatilization. In the process of the thermal volatilization of T-DMF and T-CAC, the two solvents with low boiling points (DMF and CAC) preferentially volatilized at low temperatures. However, T had less volatilization at low temperatures. Different volatilization rates result in an increased relative ratio of T to DMF and CAC after volatilization for a period, the partial pressure of T increased, and the partial pressure of DMF and CAC decreased, which caused more volatilization of T. For T-DGBE and T-DBAC, T was the component with a low boiling point and it mainly volatilized at the beginning. With the change in partial pressure, the volatilization of DGBE and DBAC increased. Therefore, mixed solvents with proper proportions can achieve hierarchical volatilization to avoid uneven volatilization. However, the adjustment ability of mixed solvents with only two components is not sufficient to meet the requirements. It is necessary to investigate the volatilization of mixed solvents with more components, and provide the flexible solvent formulation with good hierarchical volatilization.

On the basis of the volatilization results of solvents, CAC showed the most rapid volatilization and the largest standard deviation due to its low boiling point. To reduce the error of solvent volatilization, a small amount of CAC was utilized to prepare a mixed solvent for adjusting volatilization in relatively low temperatures. At the same time, T and DBAC had better volatilization stability, which made them suitable as the primary components of mixed solvents to investigate the effects of different ratios on the volatilization characteristics of mixed solvents. The volatilization characteristics of mixed solvents composed of T, DBAC, and CAC in different proportions were studied; mass fraction, volatilization curves, and amounts are shown in Figure 2 and Table 1. From the volatilization curves, the mixed solvents added with CAC showed better hierarchical volatilization, and Sample 4 showed the most obvious hierarchical volatilization characteristics. In the range of 40–130 °C, the rising range of volatilization curve was small, showing the first level of the curve. In the range of 130–170 °C, the solvent began to volatilize rapidly, and the rising range of the volatilization curve increased, showing the second level of the curve compared with the previous stage. After 170 °C, the rise of the volatilization curve was more obvious than that of the previous stage, showing the third level of the curve. In summary, the mixed solvent showed multilevel volatilization with slow volatilization at low temperatures, and rapid volatilization at high temperatures.

From Table 1, the volatilization amount of the four mixed solvents at 190 °C is more than twice that at 150 °C, and the volatilization amount of Sample 4 at 190 °C was the highest, showing commendable volatilization performance.

This was mainly due to the difference in boiling points of the three components. According to Figure 1, the volatilization of the three solvents was small at a low temperature, and the DBAC with a high boiling point was basically nonvolatile. The partial pressure of the mixed solvent was mainly concentrated in the relative content of T and DBAC, and the volatilization curve rose gently. With the temperature being close to the boiling point of CAC, CAC began to volatilize a large amount, and the volatilization curves increased rapidly. As the temperature rose to 170 °C, CAC was completely volatilized due to its small relative content, and the volatilization of the solvent was mainly concentrated in the covolatilization of T and DBAC. Compared with CAC, T and DBAC had greater relative content, so the volatilization curve showed a greater increase range. Furthermore, the relative content of T in Sample 4 was greater than that of DBAC, and the volatilization capacity of T was better than that of DBAC, so the rising range of the volatilization curve was greater, and the hierarchy was the most obvious. The volatilization ability of Sample 3 was the worst because of the highest content of DBAC in the mixed solvent. The hierarchical volatilization of the mixed solvents could thus be effectively adjusted by adjusting the proportion of the solvent components [11,14,15]. Therefore, the slurry prepared with the organic vehicle with hierarchical volatilization could achieve satisfactory discharge and sintering [16].

### 3.2. Wettability of Mixed Solvents to Glass Powder and Glass Powder

As a base metal with excellent conductivity, Cu has in recent years gradually attracted much attention in the field of electronic paste. Da Costa et al. stated that the dispersion of copper plays the most important role in sintering, and it is the main agent responsible for densification in both solid and liquid states [17]. The wettability of mixed solvent to the copper powder sheet is shown in Figure 3, which indicates that the four mixed solvents had satisfactory wettability to the copper powder sheet. Contact angles were all below 10°, which was close to complete wetting. Moreover, the wettability of the four mixed solvents to the glass powder was complete wetting (Appendix A, see the Appendix A for detailed data). According to the contact-angle data of the solvent on the metal particles and glass powder, the four mixed solvents with a mass ratio of 50:50 met the requirements of commendable wettability to the bonding and conductive phases. The better wettability of mixed solvents to copper and glass powders is beneficial to the dispersibility of particles in the slurry and the uniformity of sintering products. When the slurry is sintered, the glass powder softens, and the liquid organic vehicle can freely come into contact with the adjacent copper conductive phase. With the increase in temperature, metal particles rearrange and tend to be close. During the cooling process, the glass powder shrinks, the metal particles are in close contact, and a satisfactory conductive path is formed [18,19].

### 3.3. Solubility of Polyacrylate Resins in Mixed Solvents

Polyacrylate resin is often used as a thickener in electronic paste, the solubility of solvent to thickener is genuinely important for film-forming quality. It not only has an important impact on the rheology of organic vehicles, but also plays a certain role in the thermal degradation of polyacrylate resin; better solubility can effectively reduce the thermal degradation residue of thickeners. When the solubility parameters of solvent and thickener are closer, it is generally considered that the thickener has stronger solubility in solvents. According to Hansen’s dissolution parameter theory, the group contribution method is often used to calculate solubility parameters δ of polymers [20,21]. Hansen et al. proposed to divide the cohesive energy into three parts, i.e., $E=F_{d}+F_{p}+F_{h}$, where $F_{d}$, $F_{p}$, and $F_{h}$ are the contribution values of dispersion forces, dipole force, and hydrogen bonding, respectively. Therefore, solubility parameters can be calculated as follows:
(2)$$\delta^{2}=\delta_{d}{}^{2}+\delta_{p}{}^{2}+\delta_{h}{}^{2}$$

$\delta_{d}$ is the dispersion solubility parameter, $\delta_{p}$ is the polar solubility parameter, and $\delta_{h}$ is the hydrogen bond solubility parameter.

The three-dimensional solubility parameters of copolymers such as polyacrylate resins can be calculated by the mole fraction weighted average of homopolymer solubility parameters [22]:(3)$$\delta =\sum \delta_{i}\phi_{i}$$
where $\phi_{i}$ is the mole fraction of component *i*.

Due to the limitations of the classical model for calculating the solubility parameters of mixed solvents, Yang Long et al. [23] proposed a new nonlinear mathematical model calculation method to accurately calculate the solubility parameters of mixed solvents, and introduced the volume components of solvents as calculation parameters. The solubility parameters of mixed solvents are calculated as follows:(4)$$\delta^{2}{}_{t,m}=\psi_{1}\delta_{t,1}{}^{2}+\psi_{2}\delta_{t,2}{}^{2}+\ldots +\psi_{n}\delta_{t,n}{}^{2}$$

$\psi_{i}$ represents the volume component of component *j*, and $\delta^{2}{}_{t,m}$ represents the square of solubility parameters in each dimension.

According to the three-dimensional solubility parameters of solvents and monomers of resins in Appendix A, the solubility parameters of polyacrylate resins and mixed solvents are calculated according to Formulas (3) and (4), as shown in Table 2. The solubility parameters and intrinsic viscosity of the organic vehicles with polyacrylate resin A are shown in Figure 4.

The solubility parameters of polyacrylate resins A and B were 18.77 because their monomer compositions were the same. The solubility parameters of the four mixed solvents T-DMF, T-DBAC, T-CAC, and T-DGBE were 23.10, 19.91, 20.54, and 20.85, respectively. Among them, the solubility parameter of T-DBAC was the closest to that of resins, and the difference between the solubility parameters of T-DMF and resins was the largest, especially for the polar solubility parameter ($\delta_{p}$). When the solubility parameters of polymers are equal to the solubility parameters of solvents, the intrinsic viscosity of their solutions is maximized [24] because of the commendable thermodynamic properties. However, as shown in Figure 4, the organic vehicle of T-DMF, which had the largest difference in solubility parameters from polyacrylate resin A, showed the highest intrinsic viscosity, while the organic vehicles of T-DBAC, T-CAC, and T-DGBE conformed to the rules that the smaller the difference in solubility parameters from polyacrylate resin A is, the greater the intrinsic viscosity of organic vehicle.

DMF is a strongly polar organic solvent with a high polar solubility parameter component. Therefore, the chain stretching process of polyacrylate resin A in T-DMF is mainly controlled by the component of the polar solubility parameter. High polarity completely stretches the molecular chain, so it has larger hydrodynamic volume and greater intrinsic viscosity. Eom and Kim studied the dissolution process of polyacrylonitrile (PAN) in DMF and DMSO [25], and found that the solubility parameter of DMF (δ = 24.8) was closer to PAN (δ = 25.3) than to DMSO (δ = 26.6). However, experimental results showed that the hydrodynamic diameter of PAN and intrinsic viscosity of solution in DMSO were higher than those in DMF. This may have mainly been caused by the higher polar solubility parameter component of DMSO than that of DMF. Similarly, it also explains the behavior of the intrinsic viscosity of the organic vehicle with A dissolved in T-DMF in this paper.

For the three other mixed solvents with similar three-dimensional solubility parameters as those of polyacrylate resin A, the hydrodynamic volume of the polymer in the solvent was determined by the solvent. In the solvent that had high affinity (good solvent) for the polymer, it was easy to be dissolved, and the molecular chain was stretched, which resulted in it having large hydrodynamic volume. On the other hand, if the polymer was dissolved in a solvent with low affinity (inferior solvent), the polymer molecular chain was curled up and thus had relatively small hydrodynamic volume [26,27]. Intrinsic viscosity refers to the contribution of a single molecule to the solution viscosity. Since the hydrodynamic volume of the polymer is directly proportional to the viscosity of the polymer solution, the intrinsic viscosity of the polymer solution dissolved in a good solvent is greater than that of inferior solvents [26,27,28].

### 3.4. Analysis of Thermogravimetric Results

#### 3.4.1. Thermogravimetric Analysis of Polyacrylate Resins

To prevent the poor conductivity caused by oxidation when the copper end is sintered in oxygen atmosphere, the copper electrode MLCC is mostly sintered in a nitrogen atmosphere. In the copper-end slurry in MLCC, the incomplete thermal decomposition of a thickener in a nitrogen atmosphere causes carbon residue on the surface of the copper powder. Therefore, the selected thickener should have low residue levels in a nitrogen atmosphere. The thermogravimetric analysis results of A and B are shown in Figure 5.

The initial decomposition temperature of the sample was about 260 °C, and it was completely decomposed at about 460 °C, with the final residue rates as 1.46% and 0.21%, respectively. Comparing resin A and resin B, resin B showed a lower initial decomposition temperature and lower maximal decomposition rate temperature: $T_{max}$, was about 3 °C lower than that of A. With the same monomer compositions, the thermal decomposition stability of polymers is mainly influenced by molecular weight. The higher the molecular weight is, the worse the thermal stability, because the activation energy of polymer degradation decreases with the increase in molecular weight [29]. Meanwhile, the enhanced extraction of intermolecular hydrogen at high temperatures results in chain scission and increased thermal decomposition [30]. The longer chains showed a great tendency to form tight curves. The resulting bond angular strain led to a higher C–C scission rate [31]. In general, an increase in molecular weight increases the contribution of random fractures, resulting in different end-group structures.

The thermogravimetric curve showed two thermal decomposition peaks that were more visible in DTG curve, of which the thermal decomposition peak of near 370 °C was the most obvious. The thermal decomposition process of A and B in nitrogen atmosphere can be divided into two stages: the first stage was at 260–320 °C; the weight loss in this stage was not obvious, and it mainly involved the volatile decomposition of residual solvents and small molecular components in the polymer [32]. The second stage was at 320–460 °C; the weight loss was the most notable, and the weight loss rate reached 95%. The second degradation stage represents the degradation of A and B into small monomer molecules [33,34]. The resin produced an alkyl methacrylate monomer and a small number of other decomposition products, such as carbon monoxide, carbon dioxide, ethane, methanol, ethanol, and 1-propanol [35,36]. With the continuous fracture of the end-group structure, and the decomposition products volatilizing, the weight loss gradually increased.

#### 3.4.2. Thermogravimetric Analysis of Organic Vehicles

The organic vehicle is the key to the electronic slurry in MLCC, which plays a role in dispersing all phases and adjusting the performance of the slurry. In the process of debinding the electronic paste (150–250 °C), the organic solvent in the paste needs to have commendable volatile properties; in the sintering process of electronic paste (250–600 °C), the polyacrylate resin is required to reach better thermal degradation performance and no residue. The organic vehicle should have commendable thermal degradation performance and low residue to ensure the normal discharge process of the slurry. The TGA of organic vehicles prepared by different mixed solvents and resins was tested, and results are shown in Figure 6.

The thermal decomposition curve of the organic vehicle can be divided into two stages: the first stage was at 40–320 °C, and the weight loss rate was close to 70%. The weight loss at this stage was mainly caused by the thermal volatilization of the mixed solvents in the organic vehicles. Among them, the weight loss is also associated with the volatilization of the solvents. For both organic vehicles, the weight loss with mixed solvents of T-DMF and T-CAC started at lower temperatures because of the low boiling points of DMF and CAC. In the second stage of 320–460 °C, mixed solvents were completely volatilized, and the polyacrylate resins underwent thermal decomposition. Therefore, the weight loss rate was about 30%, which was the proportion of resins in the organic vehicles.

Comparing the thermogravimetric curves of different organic vehicles, the thermal decomposition of resin in PA-T-CAC and PA-T-DGBE began at 320 °C, the polyacrylate resins in PA-T-DMF and PA-T-DBAC had a lower thermal decomposition starting temperature, which was 10–20 °C lower than that in PA-T-CAC and PA-T-DGBE. This result indicates that T-DMF and T-DBAC reduced the thermal stability of resins A and B because the curled-chain structure of the polymer was stretched, and the hydrodynamic volume and free volume increased more in a good solvent [26,27], resulting in a reduction in the distance between molecular chains, and the decreased energy destroying the chemical bond in the molecular chain; therefore, the thermal decomposition temperature decreased to a certain extent. This shows that the intrinsic viscosity of the organic vehicle qualified as a reasonable indicator for the thermal decomposition of resin, as the intrinsic viscosity of resin A in T-DMF and T-DBAC was greater than that in T-CAC and T-DGBE. However, the solvent had no effect on the thermal stability of B. On the one hand, the solvent had a more obvious effect on the low-molecular-weight resin; on the other hand, the thermal decomposition of high-molecular-weight polymers is faster than that of a low-molecular-weight polymer.

To further verify the relationship between the intrinsic viscosity (*η*) and residual mass of resins A and B, intrinsic viscosity and mass changes Δ*m%* of the organic vehicle in the range of 320–460 °C are summarized in Figure 7. The intrinsic viscosity was directly proportional to the thermal decomposition degree of resins A and B. With the highest intrinsic viscosity, PA-T-DMF and PB-T-DMF showed the greatest degree of thermal decomposition, the mass change rate Δ*m%* was 29.58% and 29.10% (the mass ratio of A and B in the organic vehicle was 30%), and the thermal decomposition rate was 98.6% and 97.0%, respectively. PA-T-DGBE and PB-T-DGBE had the lowest intrinsic viscosity, which shows the lowest degree of thermal decomposition with the mass change Δ*m%* as 28.46%, and the thermal decomposition rate as 94.87%. Thus, the greater the intrinsic viscosity of organic vehicles is, the more completed the thermal decomposition with less residue.

## 4. Conclusions

This is an effective method to prepare mixed solvents with differential boiling points and proportions to realize the hierarchical volatilization of organics used in MLCC. Terpineol can volatilize slowly at low temperatures and rapidly at high temperatures, which is a commendable main component in mixed solvents.

The solubility parameters can be used as the theoretical basis for the selection of mixed solvents for electronic pastes, and the intrinsic viscosity can directly characterize the degree of dissolution of polymers in solvents. The closer the solubility parameters of the mixed solvent and polymer are, the greater the intrinsic viscosity of the organic vehicle is. Furthermore, intrinsic viscosity is a reasonable indicator for the thermal decomposition of resins. The thermal decomposition residue rate of organic vehicles decreases gradually with the increase in intrinsic viscosity, and the maximal thermal decomposition rate can reach 98.6%. When preparing the organic vehicle for MLCC electronic slurries, a mixed solvent with similar solubility to the used polymer can be preferentially selected in order to achieve a low carbon residue of organic vehicles in the sintering process.

## Figures and Tables

**Figure 1 polymers-14-00685-f001:**
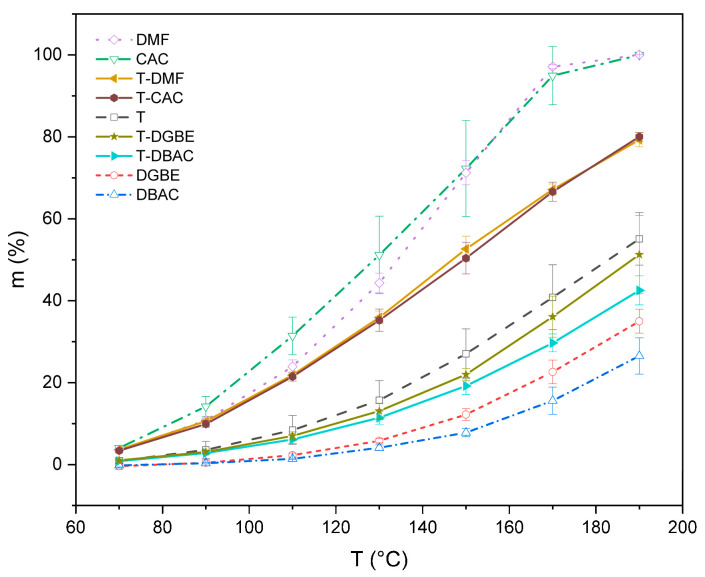
Volatilization curve of pure and mixed solvents; mean values plotted with error bars representing standard deviation.

**Figure 2 polymers-14-00685-f002:**
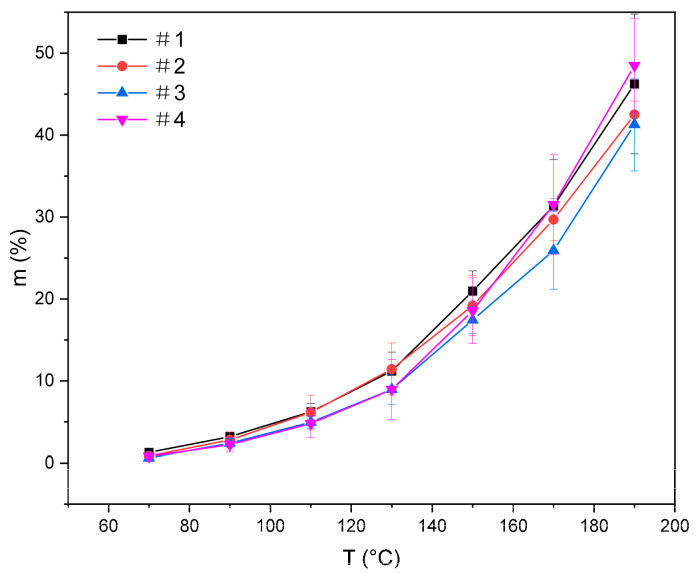
Volatilization curve of mixed solvents with various mass ratios; mean values plotted with error bars representing standard deviation.

**Figure 3 polymers-14-00685-f003:**
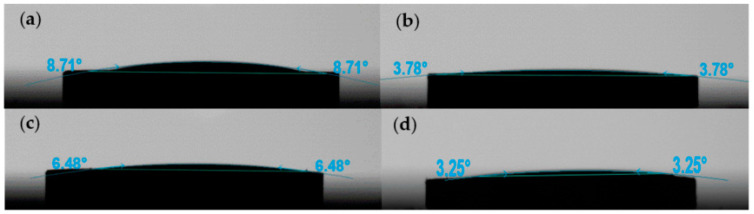
Contact angles of mixed solvents on copper powder sheet. (**a**) T-DMF; (**b**) T-DBAC; (**c**) T-DGBE; (**d**) T-CAC.

**Figure 4 polymers-14-00685-f004:**
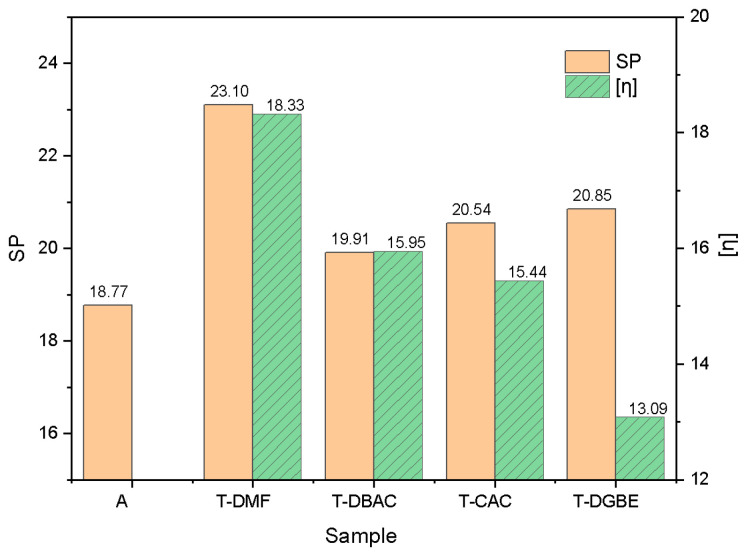
Solubility parameters (SP) and intrinsic viscosity of organic vehicles with polyacrylate resin A.

**Figure 5 polymers-14-00685-f005:**
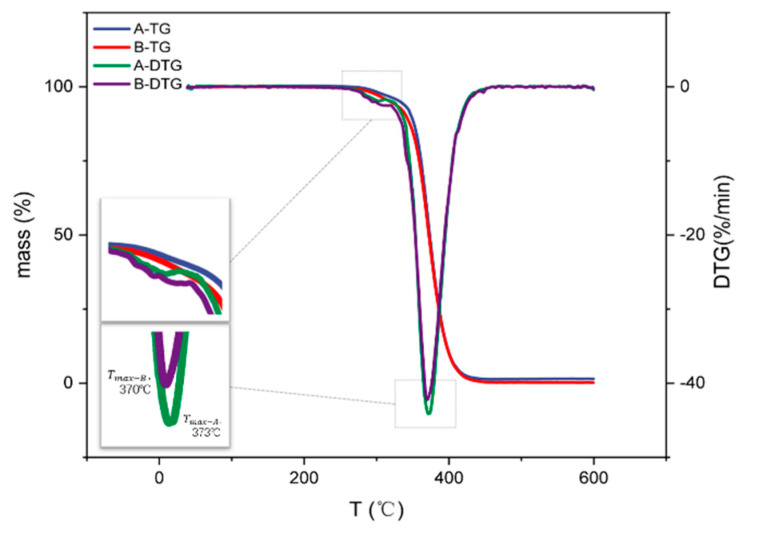
Thermogravimetric curves of polyacrylate resins A and B.

**Figure 6 polymers-14-00685-f006:**
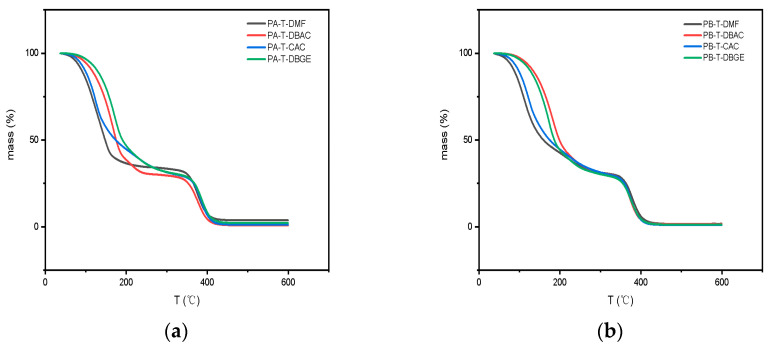
Thermogravimetric curve of organic vehicles with (**a**) resin A and (**b**) resin B.

**Figure 7 polymers-14-00685-f007:**
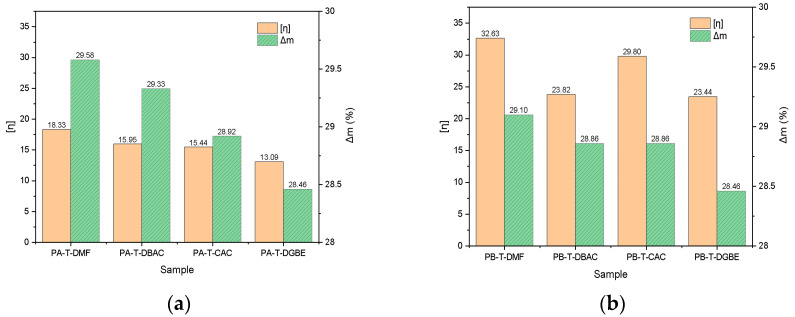
Intrinsic viscosity (*η*) and mass changes Δm of organic vehicles with (**a**) resin A and (**b**) resin B.

**Table 1 polymers-14-00685-t001:** Composition of mixed solvents and their volatilization amount at different temperatures.

Sample	T (wt/%)	DBAC (wt/%)	CAC (wt/%)	Vol (%)	
				150 °C	190 °C
1	50	40	10	20.96	46.24
2	50	50	0	19.20	42.51
3	30	60	10	17.44	41.28
4	60	30	10	18.56	48.48

**Table 2 polymers-14-00685-t002:** Three-dimensional solubility parameters of solvents and resins.

Sample	$\delta_{d}(J^{1/2}\cdot {\mathrm{cm}}^{-3/2})$	$\delta_{p}(J^{1/2}\cdot {\mathrm{cm}}^{-3/2})$	$\delta_{h}(J^{1/2}\cdot {\mathrm{cm}}^{-3/2})$	$\delta (J^{1/2}\cdot {\mathrm{cm}}^{-3/2})$
A and B	16.59	4.20	7.71	18.77
T-DMF	17.25	10.41	11.30	23.10
T-DBAC	16.56	4.87	9.93	19.91
T-CAC	16.56	5.13	11.01	20.54
T-DGBE	16.56	6.26	11.01	20.85

## Data Availability

The data presented in this study are available on request from the corresponding author.

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
