# Peer review of "Mixed Solvents in Multilayer Ceramic Capacitors (MLCC) Electronic Paste and Their Effects on the Properties of Organic Vehicle"

_polymers, 2022, doi:10.3390/polym14040685_

Round 1
Reviewer 1 Report
The authors modified the manuscript titled "Mixed solvents in multi-layer ceramic capacitors (MLCC) electronic paste and their effects on the properties of organic vehicle" according to the previous comments and I did find the overall soundness has been improved. However, significant conflicts can still be found: Figure 2 shows samples 1-4 have no difference statistically. I would therefore not recommend this manuscript to be published.
Author Response
Comments and Suggestions for Authors
The authors modified the manuscript titled "Mixed solvents in multi-layer ceramic capacitors (MLCC) electronic paste and their effects on the properties of organic vehicle" according to the previous comments and I did find the overall soundness has been improved. However, significant conflicts can still be found: Figure 2 shows samples 1-4 have no difference statistically. I would therefore not recommend this manuscript to be published.
Response: Thank you for your advice. From Figure 1, it can be seen that the mixed solvents with a mass ratio of 50:50 for Terpineol as a main component did not show good volatilization curve, which means that the mixed solvents with only two components cannot meet the requirements. Therefore, it is necessary to investigate the volatilization of mixed solvents with more components and provide the flexible solvent formulation with good hierarchical volatilization.
According to Figure 2, the four groups of samples are further discussed based on results in Figure 1, and they are all prepared by three solvents (T, CAC, DBAC) according to a certain mass ratio (detailed in Table 1.), the volatilization curves did not show an obvious difference and all solvent formulations are with good hierarchical volatilization. Strictly speaking, #4 has better hierarchical volatilization than other samples, which is characterized by less volatilization at low temperature (the volatilization amount of #4 at 90°C is 2.24%, the least of the four samples) and rapid volatilization at drying temperature (the volatilization amount of #4 at 190°C is 48.48%, the highest of the four samples). The volatilization amount of the other samples can be found in Table 1.
Reviewer 2 Report
The work «Mixed solvents in multi-layer ceramic capacitors (MLCC) electronic paste and their effects on the properties of organic vehicle» is very actual. Multi-layer ceramic capacitors have attracted great scientific and practical interest over the past years. The problem discussed in this article is unusual, but necessary in the context of the production of capacitors. The introduction succinctly reflects the field of research, its purpose and objectives. The experimental section contains a detailed description of the work, the materials and methods used. The results obtained and their explanations are accompanied by qualitative Figures. In the conclusion, the work done is correctly summed up.
One disadvantage is lack of discussion about the parameters of the polymer binder for further operations of MLCC production. It is known that lamination into multilayer coupons is an important operation after the tapes of green ceramics are cast and electrode pastes are applied to them by screen printing. Hence, the modes of polymer melting, heat shrinkage properties, and the technological mode of burnup during final annealing are of interest. The specified data can improve manuscript.
Accept after minor revision.
Author Response
Comments and Suggestions for Authors
The work «Mixed solvents in multi-layer ceramic capacitors (MLCC) electronic paste and their effects on the properties of organic vehicle» is very actual. Multi-layer ceramic capacitors have attracted great scientific and practical interest over the past years. The problem discussed in this article is unusual, but necessary in the context of the production of capacitors. The introduction succinctly reflects the field of research, its purpose and objectives. The experimental section contains a detailed description of the work, the materials and methods used. The results obtained and their explanations are accompanied by qualitative Figures. In the conclusion, the work done is correctly summed up.
One disadvantage is lack of discussion about the parameters of the polymer binder for further operations of MLCC production. It is known that lamination into multilayer coupons is an important operation after the tapes of green ceramics are cast and electrode pastes are applied to them by screen printing. Hence, the modes of polymer melting, heat shrinkage properties, and the technological mode of burnup during final annealing are of interest. The specified data can improve manuscript.
Response:
Thank you very much for your valuable suggestion! As you mentioned, the organic vehicle and its properties in sintering of MLCC production are very important. In order to meet the requirement of MLCC, the choice of the solvent and polymer and their adaptability are crucial to assure the good properties of MLCC. This manuscript focuses on the properties of the mixed solvents in MLCC mainly including the volatility of different solvents, the hierarchical volatility and solubility parameters of mixed solvents, the thermogravity of resins and organic vehicles, and the effect of solubility parameter on the dissolvability of resins in the solvent and the residual of organic vehicles. Based on the results from this work, we will further characterize the modes of polymer melting, heat shrinkage properties, and the technological mode of burnup during final annealing in the sintering process in the following work.
Reviewer 3 Report
The manuscript is well written with enough characterization results. Hene, I would accept the manuscript as it is.
Author Response
Reviewer 3
Comments and Suggestions for Authors
The manuscript is well written with enough characterization results. Hene, I would accept the manuscript as it is.
Response: Thank you very much!
Reviewer 4 Report
The paper is interesting and can be regarded useful for pursuing technical improvements in the field of capacitors. However, changes are needed before considering it fully worthy of publication. Below a list of comments and questions:
- Some more detailed information should be added in the Introduction section to underline the importance and possible applications of MLCCs.
- In section 2.2.2 please make clear that contact angle is static and say how many measurements have been taken and if values used in section 3.2 refer to an average; if so, please provide the related standard deviation.
- I suggest to clearly express the meaning Authors give to “hierarchical volatilization” the first time they mention it. It is not completely understandable.
- Authors should indicate at which temperature (or in which temperatures range) sintering occurs in order to better link such temperature to the results of TG analysis. Moreover, they said that no residual carbon species are detected (and this is fine for TG results), but this is strictly dependent on the precise temperature at which sintering is commonly performed on the real object.
- About Figure 5, no differences that could influence a practical procedure and selection of materials are really visible. Authors wrote about “two thermal decomposition peaks” (line 337), but peaks are not really present, and in general peaks can be visible in DTG, not in TG graphs of course. Please address these points and clarify the related discussion.
- Lines 343-344, Authors wrote: “The second degradation stage represents the degradation of A and B into small monomer molecules”. Do they have any experimental evidence of this?
Some typos are present in the paper, please carefully check.
Author Response
Comments and Suggestions for Authors
The paper is interesting and can be regarded useful for pursuing technical improvements in the field of capacitors. However, changes are needed before considering it fully worthy of publication. Below a list of comments and questions:
- Some more detailed information should be added in the Introduction section to underline the importance and possible applications of MLCCs.
Response: Thank you for your advice. Some detailed information about MLCC is added in line 29-36 as following:
“As one of the three basic components, the multilayer ceramic capacitor (MLCC) has the advantages of small size, high frequency and low cost compared with other components, therefore, MLCC is widely used in various electronic products and grad-ually replace traditional aluminum electrolytic capacitors, which has become one of the most widely used capacitors. At present, MLCC has been widely used in electronic equipment such as radio and television, mobile communication, measuring instruments, etc. With the continuous upgrading of electronic products such as smartphones and the popularization of new communication technologies, MLCC has a huge de-mand in the market.”
- In section 2.2.2 please make clear that contact angle is static and say how many measurements have been taken and if values used in section 3.2 refer to an average; if so, please provide the related standard deviation.
Response: Thank you. The static contact angle of the copper or glass powder sheet to the mixed solvent was tested by a contact angle tester and the measurement was repeated twice to ensure reproducibility. The data in section 3.2 are not referred to an average, they are selected random from the results obtained from the two measurements.
- I suggest to clearly express the meaning Authors give to “hierarchical volatilization” the first time they mention it. It is not completely understandable.
Response: Thanks for your advice.The ability of hierarchical volatilization requires that the mixed solvent basically does not volatilize at low temperature, and volatilizes continuously and rapidly at drying temperature. For example, as is shown in Figure 2 and Table 1, #4 has better hierarchical volatilization than other samples, which is characterized by less volatilization at low temperature (the volatilization amount of #4 at 90°C is 2.24%, the least of the four samples) and rapid volatilization at drying temperature (the volatilization amount of #4 at 190°C is 48.48%, the highest of the four samples)
To facilitate the reader’s understanding of hierarchical volatilization, we change “The organic vehicle with mixed solvent can adjust the volatilization characteristics and achieve non-volatilization at low temperature, whereas hierarchical volatilization at drying temperature” to “The organic vehicle with mixed solvent can adjust the volatilization characteristics of solvents and achieve hierarchical volatilization that non-volatilization at low temperature, whereas continuously and rapidly volatilization at drying temperature” in line 65.
- Authors should indicate at which temperature (or in which temperatures range) sintering occurs in order to better link such temperature to the results of TG analysis. Moreover, they said that no residual carbon species are detected (and this is fine for TG results), but this is strictly dependent on the precise temperature at which sintering is commonly performed on the real object.
Response: Thank you. More detailed information was added in line 388-395:
“In the process of debinding of electronic paste (150-250°C), the organic solvent in the paste needs to have commendable volatile properties; in the sintering process of electronic paste (250-600°C), the polyacrylate resin is required to reach better thermal degradation performance and no residue.”
- About Figure 5, no differences that could influence a practical procedure and selection of materials are really visible. Authors wrote about “two thermal decomposition peaks” (line 337), but peaks are not really present, and in general peaks can be visible in DTG, not in TG graphs of course. Please address these points and clarify the related discussion.
Response: Resin A and resin B are consistent in the type and proportion of monomers, the most significant difference is the molecular weight, where resin B has a molecular weight approximately twice that of A. Therefore, there is not much difference in thermal degradation performance between the two resins. we have clarified the related discussion about “two thermal decomposition peaks” in this paper, as is shown in Figure 5, the DTG curve of A and B in nitrogen atmosphere can be divided into two parts, the weight loss in the first part is not obvious or even can be ignored, and it is mainly caused by the volatile decomposition of residual solvents and small molecular components in the resins; the second weight loss part represents the degradation of resins into gas or other products, similarly, we can observe a nearly plummet of the TG curve, it corresponds to the second weight loss part of the DTG curve. Conversely, the first weight loss part of the DTG curve almost cannot be observed in the TG curve.
- Lines 343-344, Authors wrote: “The second degradation stage represents the degradation of A and B into small monomer molecules”. Do they have any experimental evidence of this?
Response: We are very sorry that we do not have any experimental evidence of “the degradation of A and B into small monomer molecules” so far. Some work has done in order to clarify the products caused by the degradation of resins, but we found that there is lack of some necessary instruments to finish this work, so we cannot provide effective data to confirm this sentence. On the other hand, we check out references about the degradation of PMMA and other polyacrylate resins which were added in the manuscript, they are consistent in the thermal degradation products of resins, such as carbon monoxide, carbon dioxide, ethane, methanol, and ethanol, to some extent this words that “the degradation of A and B into small monomer molecules” is widely accepted by many scholars.
- Some typos are present in the paper, please carefully check.
Response: After checking the paper, we made the following changes:
- Line 37, “multi-layer ceramic capacitors (MLCC)” change to “MLCC”.
- Line 59, “on the basis of” change to “based on”.
- Line 106, line 129, “mark” change to “marked”.
- Line 113, “Scentific” change to “Scientific”, “Swenden” change to “Sweden”.
- Line 122, delete “for”.
- Line 143, “thickener” change to “thickeners”.
- Line 148, delete “the”
- Line 162, “point” change to “points”.
- Line 189, 192, 327, 398, and 414, “in order to” change to “to”.
- Line 191, “rapidest” change to “the most rapid”.
- Line 194, “relative” change to “relatively”.
- Line 210, “ratio” change to “ratios”.
- Line 254, “T-CAC” change to “T-DGBE”, “T-DGBE” change to “T-CAC”.
- Line 266, “following” change to “follows”.
- Line 311, “provides an explain for” change to “explains”.
- Line 357, “amount” change to “number”.
- Line 362, “of” change to “to”.
- Line 415 and 416, “temperature” change to “temperatures”.
- Line 423, “vehicle” change to “vehicles”.
Round 2
Reviewer 1 Report
The authors have resolved my concerns. This manuscript is recommended to be published in the present version.
Reviewer 4 Report
My questions and comments have been successfully addressed.